# Primary producers may ameliorate impacts of daytime $CO_2$ addition in a coastal marine ecosystem

Matthew E.S. Bracken[1], Nyssa J. Silbiger[1,2], Genevieve Bernatchez[1] and Cascade J.B. Sorte[1]

[1] Department of Ecology and Evolutionary Biology, University of California, Irvine, Irvine, CA, United States of America
[2] Department of Biology, California State University, Northridge, Northridge, CA, United States of America

## ABSTRACT

Predicting the impacts of ocean acidification in coastal habitats is complicated by bio-physical feedbacks between organisms and carbonate chemistry. Daily changes in pH and other carbonate parameters in coastal ecosystems, associated with processes such as photosynthesis and respiration, often greatly exceed global mean predicted changes over the next century. We assessed the strength of these feedbacks under projected elevated $CO_2$ levels by conducting a field experiment in 10 macrophyte-dominated tide pools on the coast of California, USA. We evaluated changes in carbonate parameters over time and found that under ambient conditions, daytime changes in pH, $pCO_2$, net ecosystem calcification ($NEC$), and $O_2$ concentrations were strongly related to rates of net community production ($NCP$). $CO_2$ was added to pools during daytime low tides, which should have reduced pH and enhanced $pCO_2$. However, photosynthesis rapidly reduced $pCO_2$ and increased pH, so effects of $CO_2$ addition were not apparent unless we accounted for seaweed and surfgrass abundances. In the absence of macrophytes, $CO_2$ addition caused pH to decline by ~0.6 units and $pCO_2$ to increase by ~487 $\mu$atm over 6 hr during the daytime low tide. As macrophyte abundances increased, the impacts of $CO_2$ addition declined because more $CO_2$ was absorbed due to photosynthesis. Effects of $CO_2$ addition were, therefore, modified by feedbacks between $NCP$, pH, $pCO_2$, and $NEC$. Our results underscore the potential importance of coastal macrophytes in ameliorating impacts of ocean acidification.

## INTRODUCTION

Increased concentrations of $CO_2$ in the atmosphere have already altered oceanic carbonate chemistry, resulting in a decline in overall pH by ~0.1 units since the year 1800 (*Sabine et al., 2004*; *Orr et al., 2005*). As atmospheric $CO_2$ continues to increase, pH changes are predicted to accelerate, leading to an additional decline of 0.07 (RCP2.6, "high mitigation" scenario) to 0.33 (RCP8.5, "business-as-usual" scenario) pH units over the next 80 years (*Bopp et al., 2013*). Ocean acidification is predicted to dramatically affect marine ecosystems (*Harley et al., 2006*; *Doney et al., 2009*). Important resources (e.g., calcifying plankton) and

Corresponding author
Matthew E.S. Bracken,
m.bracken@uci.edu

foundation species (e.g., corals, mollusks) will have difficulty maintaining their calcium carbonate skeletons (e.g., *Orr et al., 2005*; *Pfister et al., 2016*), leading to declines in growth, abundance, and survival (*Kroeker et al., 2013*). Predicting the impacts of ocean acidification on marine communities and ecosystems is therefore critical. However, these predictions are difficult in coastal ecosystems, where carbonate chemistry is already extremely dynamic (e.g., *Wootton, Pfister & Forester, 2008*; *Thomsen et al., 2010*; *Hofmann et al., 2011*; *Guadayol et al., 2014*; *Chan et al., 2017*; *Koweek et al., 2017*; *Silbiger & Sorte, 2018*).

The range in pH values recorded in many coastal systems is greater than the pH decline predicted by the year 2100 under the RCP8.5 "business as usual" scenario (*Bopp et al., 2013*; *Hofmann et al., 2011*; *Sorte & Bracken, 2015*; *Silbiger & Sorte, 2018*). For example, coastal upwelling transports depth-derived, low-pH waters to the coast, resulting in some of the lowest pH values ever recorded in the surface ocean (*Feely et al., 2008*; *Chan et al., 2017*). Additionally, fluxes of dissolved inorganic carbon, associated with processes such as photosynthesis and respiration, are substantially higher in coastal systems than in the open ocean, leading to the suggestion that ocean acidification is an "open-ocean syndrome" (*Duarte et al., 2013*). Predicting the impacts of ocean acidification in coastal habitats is therefore complicated by the biogeochemical processes mediated by the resident biota, which drive local-scale pH variability (*Wootton, Pfister & Forester, 2008*; *DeCarlo et al., 2017*; *Silbiger & Sorte, 2018*). All organisms respire, increasing $CO_2$ concentrations and reducing pH (*Cai et al., 2011*; *Kwiatkowski et al., 2016*). Primary producers, however, also photosynthesize, drawing down $CO_2$ levels and increasing pH (*Duarte et al., 2013*; *Hendriks et al., 2014*), and natural pH variability can be exceptionally high in vegetated marine habitats such as seagrass meadows and kelp forests (*Hendriks et al., 2014*; *Kapsenberg & Hofmann, 2016*; *Koweek et al., 2017*). Both mean and maximum pH values increase with the photosynthetic capacity of primary producers (*Hendriks et al., 2014*), highlighting a potential role for submerged aquatic vegetation to reduce some of the impacts of ocean acidification in coastal ecosystems (*Delille et al., 2000*; *Semesi, Beer & Björk, 2009*; *Silbiger & Sorte, 2018*).

Effective predictions of ocean acidification impacts in coastal systems must consider the biogeochemical, oceanographic, and hydrodynamic context—including ecosystem metabolism—of these ecosystems (e.g., *Hofmann et al., 2011*; *Koweek et al., 2017*). A variety of approaches have been taken toward accomplishing this goal, including *in situ* measurements of carbonate chemistry parameters in multiple coastal ecosystems (e.g., *Hendriks et al., 2014*; *Silbiger et al., 2014*; *Kwiatkowski et al., 2016*; *Silbiger & Sorte, 2018*), characterization of natural communities associated with volcanic $CO_2$ seeps (*Fabricius et al., 2011*; *Kroeker et al., 2011*), and experimental additions of $CO_2$ in the field (*Kline et al., 2012*; *Campbell & Fourqurean, 2013*; *Sorte & Bracken, 2015*; *Brown, Therriault & Harley, 2016*; *Cox et al., 2016*). Field manipulations are an especially promising approach, but adding $CO_2$ to a dynamic coastal environment is difficult due to short water-residence times and the expense and logistical difficulty of adding regulated $CO_2$ gas (*Gattuso et al., 2014*). Here, we leverage experience from previous work developing cost-effective yeast reactors to generate $CO_2$ *in situ* in tide pools on intertidal rocky reefs (*Sorte & Bracken, 2015*)

to evaluate how the effects of $CO_2$ additions on carbonate parameters are affected by the resident biota.

We used short-term manipulations of natural tide pool habitats on the coast of northern California, USA, to evaluate the effects of increased $CO_2$ on pH and $pCO_2$ in the context of highly variable community compositions (*Bracken et al., 2008*) and carbonate dynamics (*Kwiatkowski et al., 2016*; *Silbiger & Sorte, 2018*). Our experiments were conducted during daylight hours at low tide on two concurrent days in the spring of 2016, when pools were isolated by the receding tide in the morning and remained isolated for ∼8 h. We hypothesized that impacts of $CO_2$ addition on seawater carbonate dynamics would be mediated by primary producers in the tide pools. We predicted that $pCO_2$ would decline and pH would increase over the course of our experiment due to photosynthetic fixation of $CO_2$ (*Hendriks et al., 2014*; *Kwiatkowski et al., 2016*; *Silbiger & Sorte, 2018*) and that the effects of $CO_2$ addition on pH and $pCO_2$ would diminish with increasing percent cover of primary producers in the pools (*Hendriks et al., 2014*).

## MATERIALS AND METHODS

### Study location and characteristics

We conducted a $CO_2$ addition experiment on a rocky intertidal shoreline on the northern side of Horseshoe Cove in the Bodega Marine Reserve (BMR), Sonoma County, California, USA (NAD83 38.31672, −123.0711). This work was conducted with the approval of the California Department of Fish and Wildlife (Scientific Collecting Permit number SCP-13405). Our work in the BMR was performed under Research Application # 32752 to the University of California Natural Reserves System to conduct measurements of carbonate chemistry in tide pools. At the site, we marked and surveyed 10 tide pools in the mid-intertidal zone. For each pool, we determined tidal elevation (mean $= 1.54 \pm 0.05$ [s.e.] m above mean lower-low water; range $= 1.22$ to $1.73$ m), volume (mean $= 22.3 \pm 5.0$ L; range $= 6.7$ to $47.3$ L), and bottom surface area (mean $= 0.35 \pm 0.08$ m$^2$; range $= 0.16$ to $1.00$ m$^2$). Tidal elevations were determined using a self-leveling rotary laser level (CST/berger, Watseka, Illinois, USA), with reference to published tidal predictions for Bodega Harbor Entrance (*NOAA, 2016*). Pool volumes were determined by adding 2 ml of non-toxic blue dye (McCormick, Sparks, Maryland, USA) to each pool. Water samples were read at 640 nm on a benchtop spectrophotometer (UV-1800, Shimadzu, Carlsbad, California, USA) and compared to a standard curve relating absorbance to an equivalent amount of dye added to known volumes of seawater (*Pfister, 1995*). The abundances of primary producers and sessile invertebrates in each tide pool were determined by spreading a flexible mesh grid (10 cm × 10 cm mesh; Foulweather Trawl Supply, Newport, Oregon, USA) over the bottom of each pool and measuring the cover of macrophytes (seaweeds and surfgrass) and sessile invertebrates (barnacles, mussels, sea anemones, and sponges) and the total surface area of each tide pool (*Bracken & Nielsen, 2004*) (see Table S1). Mobile invertebrates (particularly turban snails) were also counted in each pool. Tide pool volumes, primary producer abundances, and surface areas were measured immediately after our experiment to avoid disturbing the pools prior to water sampling.

## Experimental design and sampling

These pools included some of the same tide pools used in other studies of carbonate chemistry under ambient conditions: *Kwiatkowski et al. (2016)* measured calcification during the spring of 2014 and 2015, and *Silbiger & Sorte (2018)* quantified bio-physical feedbacks during the summer of 2016. Our $CO_2$ addition experiments were conducted on two consecutive days: 31 March and 01 April 2016. On those days, the receding tide isolated pools in the morning (08:30–09:30 on day 1, 09:00–10:00 on day 2), and they remained isolated for approximately 7.5 hr. Pools were randomly assigned to experimental treatments: control or $+CO_2$, with $n = 5$ pools assigned to each treatment. There were no differences in tidal elevation ($t = 1.4$, $df = 8$, $p = 0.210$), volume ($t = 0.6$, $df = 8$, $p = 0.570$), or surface area ($t = 0.8$, $df = 8$, $p = 0.453$) between control and $+CO_2$ pools. Treatments were switched between day 1 and day 2, so that pools assigned to $+CO_2$ treatments on day 1 were assigned to control treatments on day 2, and vice versa. This switching allowed us to assess the effect of $CO_2$ addition on carbonate chemistry within individual pools by quantifying the difference between day 1 and day 2. We assume that because $CO_2$ additions were within the natural variability of the system, there were minimal carryover effects of day 1 $CO_2$ additions to day 2 control tide pools. Average air temperature (day 1: $11.2 \pm 0.3\,°C$; day 2: $11.8 \pm 0.4\,°C$), wind speed (day 1: $14.7 \pm 1.3$ km hr$^{-1}$; day 2: $16.8 \pm 0.5$ km hr$^{-1}$), and photosynthetically active radiation (day 1: $871 \pm 188\ \mu$mol m$^{-2}$ s$^{-1}$; day 2: $1{,}058 \pm 173\ \mu$mol m$^{-2}$ s$^{-1}$) values, measured at the Bodega Ocean Observing Node weather station adjacent to our study site, did not differ substantially between day 1 and day 2.

$CO_2$ was delivered to $+CO_2$ pools using yeast reactors consisting of watertight plastic boxes (Drybox 2500, OtterBox, Fort Collins, Colorado, USA) containing 500 mL of warm water, 2 g of $NaHCO_3$ (to buffer the internal pH of the reactor), and 0.7 g of baker's yeast, an amount calculated to generate pH levels and $pCO_2$ concentrations that were within the predicted range of 21st-century ocean acidification scenarios (*Bopp et al., 2013*). Tubing from each reactor led to an air stone anchored within each $+CO_2$ tide pool. $CO_2$ is the most abundant gas produced by baker's yeast, representing $\sim$99.2% of headspace volume (*Daoud & Searle, 1990*), though additional volatile organic compounds such as ethanol, 2-methylpropanol, and ethyl acetate are present in low concentrations (*Smallegange et al., 2010*).

We calibrated the yeast reactors and determined the amount of yeast to use in our experiment by running a series of trials in buckets across a range of temperatures and yeast amounts. Buckets ($n = 24$) contained 12 L of saltwater at typical ocean concentrations (Instant Ocean® Sea Salt, Blacksburg, Virginia, USA). Temperature was controlled by placing buckets in growth chambers (Percival Scientific, Perry, Iowa, USA) set to 4°, 10°, 20°, or 30 °C. We measured pH (total scale) in each bucket every hr for 6 hr and monitored temperature using TidbiT® v2 dataloggers (Onset Computer Corporation, Bourne, Massachusetts, USA). These trials also represent controls, allowing us to evaluate the effects of yeast and temperature on tide pool pH in the absence of organisms.

Seawater chemistry and other environmental conditions were measured throughout the low tide period on each day. Water samples were collected from pools every 1.5 hr for

6 hr, beginning as soon as each pool was isolated by the receding tide. However, due to nonlinearity in the carbonate parameters across time—potentially due to carbon limitation as $p\mathrm{CO_2}$ declined over time (*Maberly, 1990*)—we calculated rates of change in pH, $p\mathrm{CO_2}$, net community production, and net ecosystem calcification using only the initial and final samples for each pool, which represented the overall net change over the sampling period. Temperature, dissolved oxygen and conductivity were simultaneously measured in each pool using sensors (ProODO and Pro30, YSI, Yellow Springs, Ohio, USA). Samples were collected by pumping water (400 mL) into a plastic Erlenmeyer flask using a separate piece of tubing anchored in each pool to minimize disturbance and off-gassing. Anchoring the tubing in each pool also ensured that our samples were collected from the same location every time and that they were not collected adjacent to the airstones delivering $\mathrm{CO_2}$ to $+\mathrm{CO_2}$ pools. Each water sample was immediately divided into subsamples for analyses of pH, total alkalinity, and dissolved inorganic nutrients. Prior to sample collection, all containers were washed in 10% HCl, rinsed 3× with DI water, and rinsed 3× with ocean water. We measured pH by measuring voltage (mV) and temperature (°C) of a 50 mL subsample immediately after collection using a multiparameter pH meter (Orion Star, Thermo Fisher Scientific, Waltham, Massachusetts, USA) with a ROSS Ultra glass electrode (Thermo Scientific, USA; accuracy ± 0.2 mV, resolution ± 0.1, drift <0.005 pH units per day) and a traceable digital thermometer (5-077-8, accuracy = 0.05°C, resolution = 0.001 °C; Control Company, Friendswood, TX, USA). pH (total scale) was calculated using a multipoint calibration to a Tris standard (Marine Physical Laboratory, Scripps Institution of Oceanography, La Jolla, California, USA), as described in SOP 6a (*Dickson, Sabine & Christian, 2007*). Subsamples for determination of total alkalinity were fixed with 100 µL of 50% saturated $\mathrm{HgCl_2}$ and stored in 250 mL brown HDPE bottles. Subsamples for dissolved inorganic nutrients ($\mathrm{NO_3^-}$, $\mathrm{NO_2^-}$, $\mathrm{NH_4^+}$, and $\mathrm{PO_4^{3-}}$) were filtered (GF/F, Whatman, Maidstone, UK) into 50 ml centrifuge tubes and frozen at $-20$ °C prior to analyses.

## Sample processing

Subsamples for total alkalinity were analyzed using open-cell titrations (T50, Mettler-Toledo AG, Schwerzenbach, Switzerland), as described in SOP 3b (*Dickson, Sabine & Christian, 2007*). A certified reference material (Marine Physical Laboratory, Scripps Institution of Oceanography, La Jolla, California, USA) was run daily. Our measurements of the reference material never deviated more than ±0.4% from the certified value, and alkalinity calculations were corrected for these deviations. $\mathrm{NO_3^-} + \mathrm{NO_2^-}$ and $\mathrm{PO_4^{3-}}$ concentrations ($\mu$mol $\mathrm{L^{-1}}$) were measured on a QuickChem 8500 Series Analyzer (Lachat Instruments, Loveland, Colorado, USA), and $\mathrm{NH_4^+}$ concentrations ($\mu$mol $\mathrm{L^{-1}}$) were measured using the phenolhypochlorite method (*Solórzano, 1969*) on a UV-1800 benchtop spectrophotometer (Shimadzu, Carlsbad, California, USA). *In situ* pH values and other carbonate parameters were calculated using the *seacarb* package in *R* v. 3.2.2 (R Foundation for Statistical Computing, Vienna, Austria) (*Gattuso et al., 2017*) (Table S2). We note that error propagation for calculating $\Omega_{\mathrm{arag}}$ based on pH and TA is ~3.6% (*Riebesell et al., 2011*).

## Calculation of net community production and net ecosystem calcification in control tide pools

Differences in dissolved inorganic carbon ($DIC$) were used to calculate net community production ($NCP$) rates (mmol C m$^{-2}$ hr$^{-1}$), as described in *Gattuso, Frankignoulle & Smith (1999)*:

$$NCP = \frac{\Delta DIC \cdot \rho \cdot V}{SA \cdot t} - NEC - FCO_2$$

where $\Delta DIC$ is the difference in salinity-normalized $DIC$ between the first and last time points (in this case 0 and 6 hr; mmol kg$^{-1}$), $\rho$ is the density of seawater (1,023 kg m$^{-3}$), $V$ is the volume of water in the tide pool at each time point, $SA$ is the bottom surface area of each pool (m$^2$), and $t$ is the time interval between sampling points (6 hr).

$NEC$ is the rate of net ecosystem calcification (mmol CaCO$_3$ m$^{-2}$ hr$^{-1}$), which was calculated as follows:

$$NEC = \frac{\Delta TA \cdot \rho \cdot V}{2 \cdot SA \cdot t}$$

where, in addition to variables defined above, $\Delta TA/2$ is the difference in total alkalinity ($TA$, mmol kg$^{-1}$) between the time points. $TA$ was normalized to a constant salinity and corrected for dissolved inorganic nitrogen and phosphorus to account for their small contributions to the acid–base system (*Wolf-Gladrow et al., 2007*), including the potential for primary producers to change alkalinity via nutrient uptake (*Brewer & Goldman, 1976*; *Stepien, Pfister & Wootton, 2016*). One mole of CaCO$_3$ is formed per two moles of $TA$, hence the divisor of 2.

Finally, $FCO_2$, the air-sea flux of CO$_2$ (mmol m$^{-2}$ hr$^{-1}$), was calculated as follows:

$$FCO_2 = k \cdot s \cdot (CO_{2[water]} - CO_{2[air]})$$

where $k$ is the gas transfer velocity (m hr$^{-1}$), and $s$ is the solubility of CO$_2$ in seawater, which was calculated based on *in situ* measurements of temperature and salinity (*Weiss, 1974*). The concentration of CO$_2$ in air was assumed to be 400 μatm (*Tans & Keeling, 2017*). The transfer velocity of CO$_2$ was based on wind velocities measured at the Bodega Ocean Observing Node weather station located 100 m from our study location. Calculated $FCO_2$ values (mmol kg$^{-1}$ hr$^{-1}$) were converted to mmol CO$_2$ m$^{-2}$ hr$^{-1}$ based on the volume of the tide pool, the density of seawater, and the bottom surface area. All data for $NCP$, $NEC$, and $FCO_2$ are provided in the electronic supplementary materials (Table S3).

## Statistical analyses

Changes in pH in buckets were evaluated as a function of temperature (°C) and yeast (g) using multiple linear regression (PROC GLM) in SAS v. 9.4 (SAS Institute, Cary, North Carolina, USA). Changes in pH and $pCO_2$ in field tide pools over time (i.e., pH units hr$^{-1}$ or μatm hr$^{-1}$) were calculated by subtracting the initial value (0 hr) from the final value (6 hr) and dividing by the elapsed time. The difference between +CO$_2$ and control pools was evaluated for each individual pool (the experimental unit in all analyses). We quantified the effect of macrophytes (seaweeds and surfgrass) on carbonate parameters by calculating the cover of non-calcifying macrophytes in each tide pool. We excluded
calcifying species because of low photosynthetic biomass relative to non-calcifying species from this location (*Bracken & Williams, 2013*). On average, less than 5% of algal cover in the pools was composed of calcifying seaweeds. We also quantified the effect of invertebrates by estimating total invertebrate cover based on our surveys. For mobile invertebrates (mostly turban snails), we estimated cover from count data based on the number of individuals of each species in a 10 cm × 10 cm quadrat ($=0.01$ m$^2$). We then evaluated relationships between macrophyte and invertebrate abundances and net community production (*NCP*) using linear regression (PROC GLM) in SAS. To account for the effects of primary producers on pH and $p$CO$_2$, we used linear regression to evaluate the difference between control and +CO$_2$ tide pools as a function of the abundance of macrophytes in the pools. We calculated these differences by subtracting the rates of change in pH (total scale hr$^{-1}$) or $p$CO$_2$ ($\mu$atm hr$^{-1}$) in each pool under ambient CO$_2$ conditions from the rates when CO$_2$ was added to those same pools. Over time, pH increased and $p$CO$_2$ decreased due to photosynthesis. Thus, a *negative* effect of CO$_2$ addition on pH indicated that CO$_2$ addition reduced pH in +CO$_2$ pools relative to control pools. Conversely, a *positive* effect of CO$_2$ addition on $p$CO$_2$ indicated that CO$_2$ addition enhanced $p$CO$_2$ in +CO$_2$ pools relative to control pools. We assessed how pH, $p$CO$_2$, net ecosystem calcification (*NEC*), and O$_2$ responded to changes in *NCP* under ambient CO$_2$ conditions using linear regression (PROC GLM in SAS). Similarly, we evaluated whether *NEC* was related to cover of calcifiers (mussels, turban snails, and coralline algae) using linear regression. Prior to running linear regressions, we verified that data met the assumptions of normality and homogeneity of variances.

## RESULTS

Biological processes resulted in substantial changes in both pH and $p$CO$_2$ in tide pools over the course of a single low tide: pH increased by ∼0.4 units and $p$CO$_2$ decreased by ∼233 $\mu$atm in both control and +CO$_2$ pools (Fig. 1; Table S4). The effects of our experimental CO$_2$ additions on pH and $p$CO$_2$ were masked by the dominant effects of primary producers on carbonate chemistry; there was no apparent difference in either the pH ($t = 0.6$, $df = 9$, $P = 0.591$; Fig. 1A) or the $p$CO$_2$ ($t = 0.3$, $df = 9$, $P = 0.777$; Fig. 1B) of control versus +CO$_2$ pools. Physical parameters –such as changes in temperature ($R^2 = 0.17$, $F_{1,8} = 1.6$, $P = 0.238$) and light availability ($R^2 < 0.01$, $F_{1,8} < 0.1$, $P = 0.996$)—had minimal effects, by themselves, on changes in tide pool pH.

Effects of CO$_2$ additions on pH and $p$CO$_2$ only became apparent after accounting for primary producer abundances in the tide pools (Fig. 2). As the abundance of macrophytes (seaweeds and surfgrass) increased, the negative effect of CO$_2$ addition on pH was ameliorated ($R^2 = 0.45$; $F_{1,8} = 6.6$, $P = 0.033$; Fig. 2A), indicating that macrophytes limited the reduction in pH associated with CO$_2$ addition. The effect of CO$_2$ addition on pH in the absence of non-calcifying macrophytes is represented by the $y$-intercept of this relationship (Fig. 2A), which indicates a reduction in pH of 0.07 ($\pm 0.03$ s.e.) units hr$^{-1}$ ($t = 2.1$, $df = 9$, $P = 0.062$). We compared this value to the change in pH predicted by our bucket calibration trials. In the absence of organisms, pH declined more rapidly as

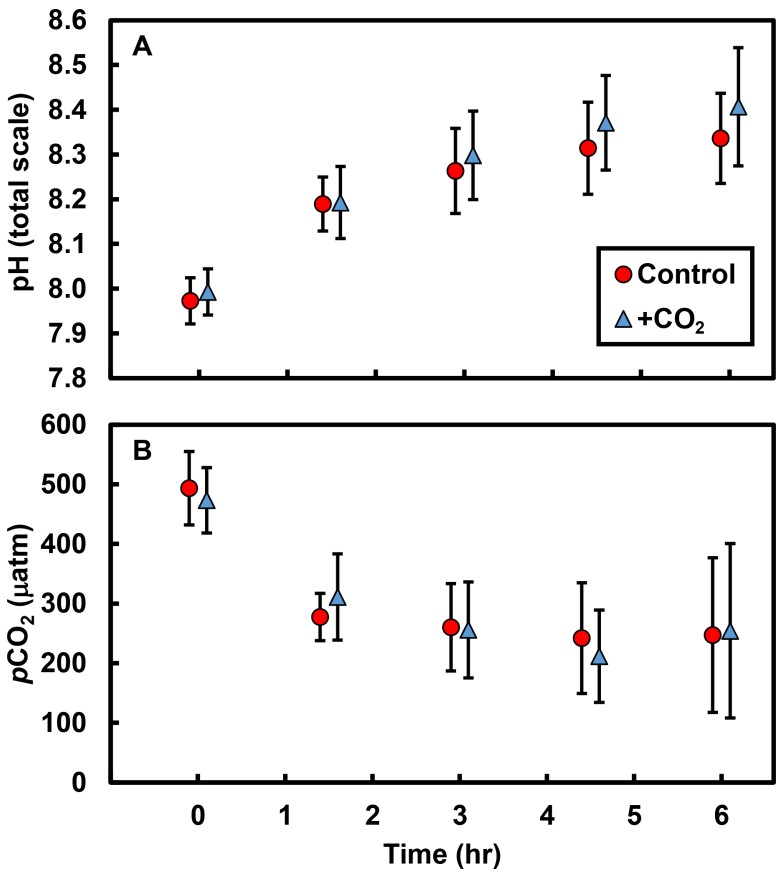

**Figure 1 Tide pool pH and $pCO_2$ values measured over time.** Over 6 hr following isolation of pools by the receding tide, (A) pH increased by an average of 0.4 units (total scale) and (B) $pCO_2$ declined by an average of ~233 μatm. However, there were no differences in pH ($P = 0.591$) or $pCO_2$ ($P = 0.777$) between control pools (with no $CO_2$ added; red circles) and +$CO_2$ pools (blue triangles). Values are means ±s.e.

temperature ($F_{1,21} = 20.1, P < 0.001$) and yeast amount ($F_{1,21} = 44.2, P < 0.001$) increased. We used the parameter estimates from the multiple linear regression, the amount of yeast (0.7 g), and the ambient tide pool water temperature (17.7 °C) to predict the change in pH in the absence of organisms. We predicted a reduction in pH of 0.06 ($\pm 0.03$ s.e.) units hr$^{-1}$, which is similar to the observed rate in the absence of macrophytes ($t = 0.2, df = 32, P = 0.845$). Correspondingly, as macrophyte cover increased in tide pools, the effect of $CO_2$ addition on $pCO_2$ was reduced ($R^2 = 0.42; F_{1,8} = 6.0, P = 0.040$; Fig. 2B). The $y$-intercept of this relationship (Fig. 2B) represents the effect of $CO_2$ addition on $pCO_2$ in the absence of non-calcifying macrophytes and indicates an increase of 81.22 ($\pm 33.80$ s.e) μatm hr$^{-1}$ ($t = 2.4, df = 9, P = 0.041$).

The effects of macrophytes on pH and $pCO_2$ emerged because increases in macrophyte abundance were associated with increases in net community production (*NCP*, mmol C m$^{-2}$ hr$^{-1}$; Fig. 3A). This relationship held for all non-calcifying macrophytes considered together ($R^2 = 0.47; F_{1,8} = 7.1, P = 0.029$) and was especially apparent for the most abundant macrophyte in the tide pools, the red seaweed *Prionitis sternbergii* (C. Agardh)

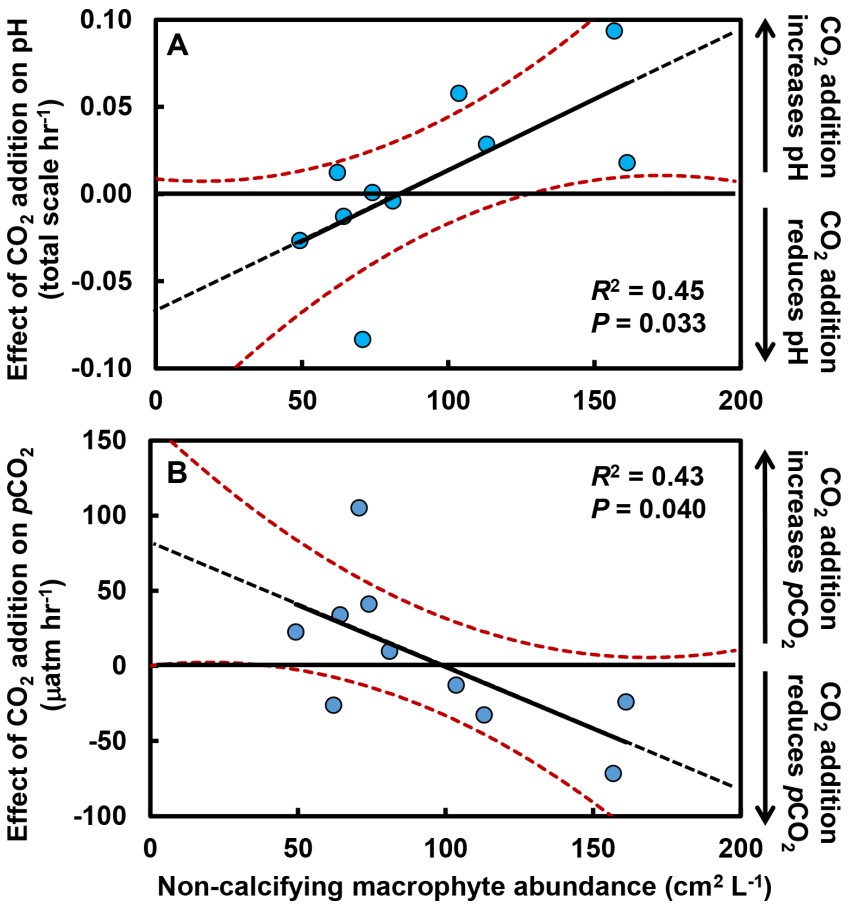

**Figure 2  Effects of CO$_2$ addition on rates of change in pH (total scale hr$^{-1}$) and $p$CO$_2$ ($\mu$atm hr$^{-1}$) in tide pools were reduced as macrophyte cover increased.** The effect of CO$_2$ addition was defined for each pool as the rate of change when CO$_2$ was added minus the rate when CO$_2$ was at ambient levels. As macrophyte abundance in the pools increased, (A) the effect of CO$_2$ addition on pH became more positive ($R^2 = 0.45$, $P = 0.033$) and (B) the effect of CO$_2$ addition on $p$CO$_2$ became more negative ($R^2 = 0.43$, $P = 0.040$). Rates of change in pH and $p$CO$_2$ in the absence of macrophytes are indicated by the $y$-intercept of the regression lines (pH:$-0.07 \pm 0.03$ s.e. total scale hr$^{-1}$; $p$CO$_2$: $+81.22 \pm 33.80$ $\mu$atm hr$^{-1}$). Regression lines are means $\pm95\%$ c.i.

J. Agardh ($R^2 = 0.68$; $F_{1,8} = 16.8.1$, $P = 0.004$). Non-calcifying macrophytes, including the seaweeds *Gelidium coulteri* Harvey, *Hildenbrandia rubra* Sommerfelt (Meneghini), *Mastocarpus papillatus* (C. Agardh) Kützing (both upright and *Petrocelis* forms), *Mazzaella splendens* (Setchell & N. L. Gardner) Fredericq, and *P. sternbergii* and the surfgrass *Phyllospadix torreyii* S. Watson, were abundant in the pools, collectively composing 56% of cover on the benthos (Fig. 3B). However, *NCP* was not related to cover of any macrophyte species other than *Prionitis* (e.g., *Hildenbrandia* [$R^2 = 0.18$; $F_{1,8} = 1.8$, $P = 0.216$], *Phyllospadix* [$R^2 = 0.0.1$; $F_{1,8} < 0.1$, $P = 0.846$]). *NCP* was also unrelated to abundances of invertebrates (primarily mussels [*Mytilus californianus* T. A. Conrad], sea anemones [*Anthopleura* spp.], and turban snails [*Tegula funebralis* A. Adams]) in the tide pools ($R^2 = 0.09$; $F_{1,8} = 0.8$, $P = 0.409$). Net ecosystem calcification (*NEC*) was unrelated

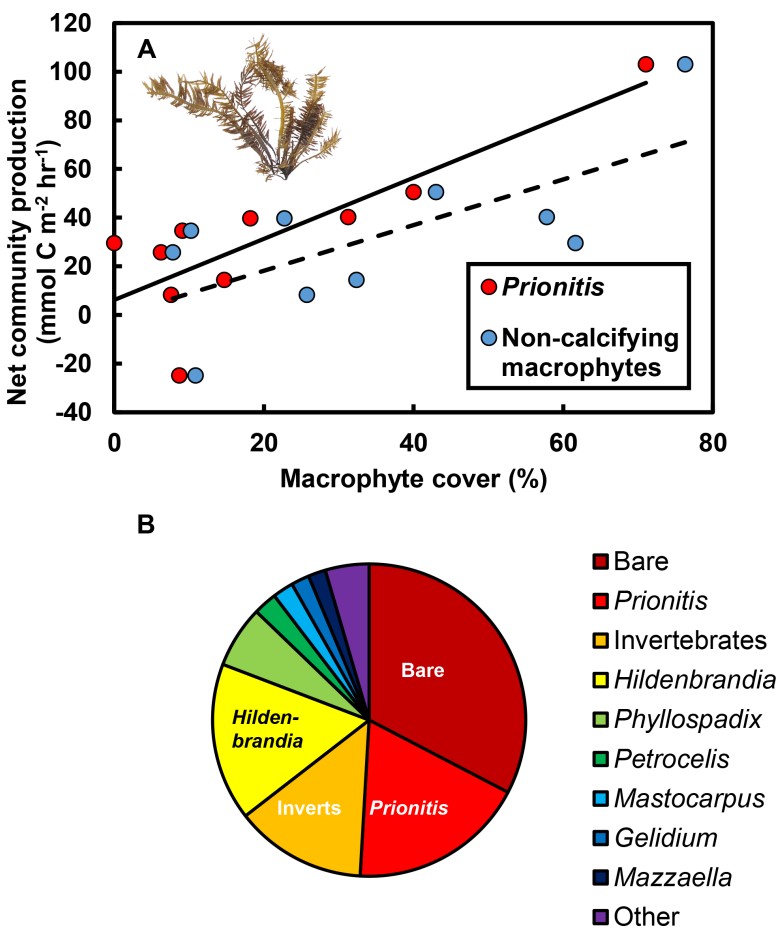

**Figure 3   Effects of macrophytes on net community production ($NCP$, mmol C m$^{-2}$ hr$^{-1}$).** (A) Increases in macrophyte cover were associated with increases in $NCP$ for both the most abundant macrophyte, the red alga *Prionitis sternbergii* (red circles, solid trendline; $R^2 = 0.68$, $P = 0.004$), and all macrophytes (blue triangles, dashed trendline; $R^2 = 0.47$, $P = 0.029$). Inset image: *P. sternbergii*. Image credit: Matthew Bracken. (B) Macrophytes, including seaweeds and the surfgrass *Phyllospadix*, dominated cover in the pools, collectively occupying the majority of the substratum. Macrophytes in the "Other" category each contributed <5% of cover and included *Cladophora*, *Endocladia*, *Mazzaella flaccida*, and calcifying species. Invertebrates represented ∼15% of total cover.

to abundances of calcifying invertebrates in the tide pools, including *Mytilus* ($R^2 = 0.03$; $F_{1,8} = 0.2$, $P = 0.653$), *Tegula* ($R^2 = 0.03$; $F_{1,8} = 0.2$, $P = 0.654$), and coralline algae ($R^2 < 0.01$; $F_{1,8} = 0.1$, $P = 0.788$).

Rates of change in pH ($R^2 = 0.63$; $F_{1,8} = 13.7$, $P = 0.006$; Fig. 4A), $pCO_2$ ($R^2 = 0.43$; $F_{1,8} = 6.1$, $P = 0.039$; Fig. 4B), and $NEC$ ($R^2 = 0.67$, $F_{1,8} = 16.2$, $P = 0.004$; Fig. 4C) were strongly related to changes in $NCP$. O$_2$ production in the tide pools was also strongly related to $NCP$ ($R^2 = 0.65$, $F_{1,8} = 14.7$, $P = 0.005$; Fig. 4D), linking macrophyte abundances (Fig. 3A), $NCP$ (measured by evaluating carbonate parameters), and O$_2$ production.

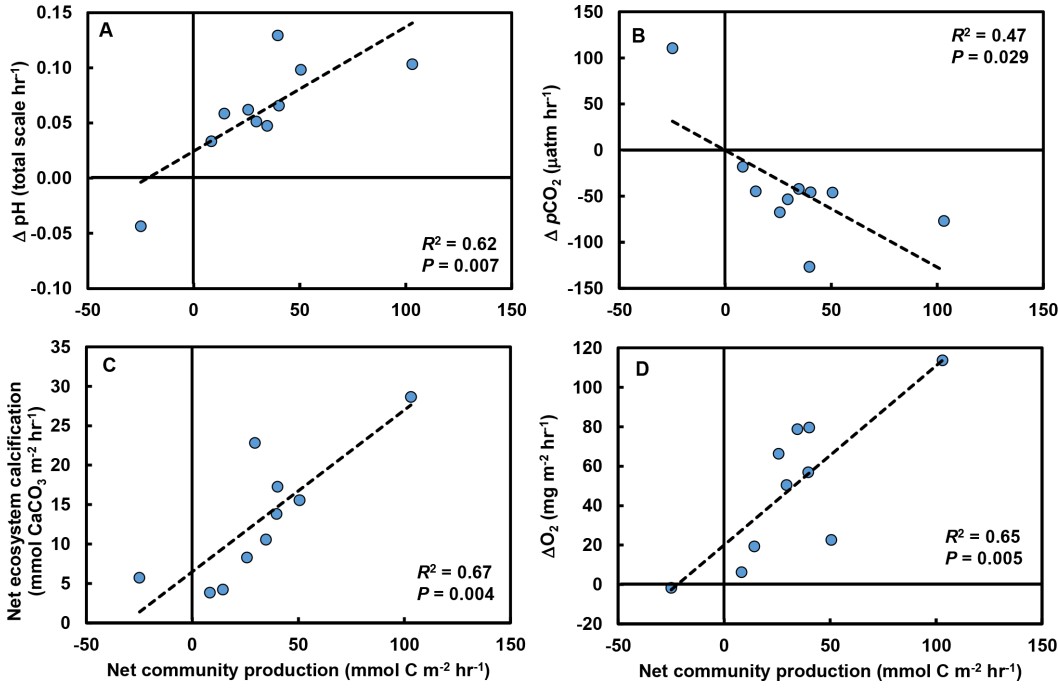

**Figure 4** **Relationships between net community production ($NCP$) and other carbonate parameters in tide pools.** Increases in $NCP$ (mmol C m$^{-2}$ hr$^{-1}$) in control tide pools were associated with (A) increases in pH (total scale hr$^{-1}$; $R^2 = 0.62$, $P = 0.007$), (B) declines in $pCO_2$ (μatm hr$^{-1}$; $R^2 = 0.47$, $P = 0.029$), (C) increases in net ecosystem calcification (mmol CaCO$_3$ m$^{-2}$ hr$^{-1}$; $R^2 = 0.67$, $P = 0.004$), and (D) increases in O$_2$ concentrations (mg m$^{-2}$ hr$^{-1}$; $R^2 = 0.65$, $P = 0.005$).

## DISCUSSION

Primary producers modified the effects of $CO_2$ addition on pH and $pCO_2$ in the tide pools, ameliorating the negative effect of $CO_2$ addition on pH and reducing the effect of $CO_2$ addition on $pCO_2$. In fact, at the highest macrophyte abundances, the effect of $CO_2$ addition on the change in pH was positive, indicating that added $CO_2$ may be ameliorating carbon limitation and enhancing rates of photosynthesis (e.g., *Gao et al., 1991*)—and thereby amelioration of low pH conditions—during the day. Based on rates of change in pH and $pCO_2$ in the absence of macrophytes, by the time pools were re-submerged by the rising tide, our yeast reactors had, in effect, elevated $pCO_2$ by ~487 μatm relative to control pools, reducing pH by ~0.4 units over ~6 hr. Observed effects of net community production ($NCP$) on net ecosystem calcification ($NEC$) highlight the potential for primary producers to not only mediate pH and $pCO_2$, but to also enhance other ecosystem functions during the day. Furthermore, the relationship between O$_2$ production and $NCP$ suggests that $CO_2$ drawdown in the tide pools was due to photosynthesis.

Our results underscore the difficulty of quantifying the effects of realistic levels of $CO_2$ addition on pH and $pCO_2$ in a producer-dominated coastal ecosystem and suggest that macrophytes such as seagrasses and seaweeds have the potential to ameliorate some impacts of ocean acidification in marine systems (*Delille et al., 2000*; *Frieder et al., 2012*;

*Hendriks et al., 2014*). As shown in previous research (*Delille et al., 2000*; *Hendriks et al., 2014*), including measurements from many of the tide pools included in the current study (*Kwiatkowski et al., 2016*; *Silbiger & Sorte, 2018*), photosynthetic uptake of $CO_2$ plays a dominant role in driving carbonate chemistry in coastal marine systems, especially during the daytime (*Duarte et al., 2013*). For example, we found that *NCP* was closely coupled to changes in pH, $pCO_2$, and *NEC*. The fact that macrophyte-mediated enhancement of production also enhanced calcification is particularly important in the context of impacts of ocean acidification on calcifying organisms.

One important caveat, however, is that our experiments were only conducted during daylight hours, when photosynthetic draw-down of $CO_2$ is maximized. Thus, we have no data to address whether macrophytes can ameliorate impacts of $CO_2$ additions at night, when respiration adds $CO_2$ and contributes to dissolution of calcifying species (*Kwiatkowski et al., 2016*; *Silbiger & Sorte, 2018*). Can daytime reduction of $pCO_2$ and enhancement of *NEC* by primary producers help to counterbalance nighttime dissolution? Several lines of evidence suggest that the answer may be yes. First, the enhancement of calcification during daylight hours is substantially greater than dissolution at night in tide pools at our study site, suggesting that producers can counterbalance pH-mediated declines in net calcification (*Kwiatkowski et al., 2016*). Second, research in seagrass beds indicates that both maximum pH values and mean pH values —averaged across both daytime and nighttime measurements— increase as macrophyte abundance increase, demonstrating a dominant effect of macrophyte abundance on pH overall (*Hendriks et al., 2014*). Third, photosynthesis, growth, and calcification can be higher under fluctuating pH conditions than under stable conditions (*Dufault et al., 2012*; *Britton et al., 2016*; *Price et al., 2012*). For example, mussels have been shown to shift the timing of shell production to daylight hours when macrophytes ameliorate impacts of elevated $CO_2$ (*Wahl et al., 2018*). Finally, whereas daytime pH values are strongly related to producer dominance in tide pools, nighttime pH is unrelated to community composition (*Silbiger & Sorte, 2018*). Higher *NCP* during the day does not translate to higher community respiration at night, and overall pH values are higher in pools dominated by primary producers.

Tide pools have long served as model systems for evaluating ecological and biogeochemical processes in coastal habitats, as their isolation at low tide allows manipulation and measurement of replicated local ecosystems (e.g., *Lubchenco, 1978*; *Dethier, 1984*; *Pfister, 1995*; *Bracken & Nielsen, 2004*; *Bracken et al., 2008*; *Sorte & Bracken, 2015*). However, they also represent an extreme case, as they are physically isolated from the surrounding ocean during low tide. Although our local-scale $CO_2$ additions only effectively manipulated pH while pools were isolated, there is some evidence, based on research in subtidal kelp beds and seagrass meadows, that marine macrophytes can substantially alter nearshore carbonate chemistry and potentially limit impacts of ocean acidification in more open nearshore systems (*Delille et al., 2000*; *Frieder et al., 2012*; *Hendriks et al., 2014*; *Nielsen et al., 2018*).

## CONCLUSIONS

In conclusion, we have shown that primary producers can mediate daytime carbonate chemistry in a coastal ecosystem and that the effects of productivity on pH and $p\text{CO}_2$ can mask the effects of short-term $\text{CO}_2$ addition during daylight hours. Based on these results, we suggest that primary producers, especially in highly vegetated coastal systems, have the potential to reduce impacts of increasing $\text{CO}_2$ concentrations in those systems. However, large, dominant macrophytes, including kelps and seagrasses, that have the potential to ameliorate the impacts of ocean acidification (*Frieder et al., 2012*; *Hendriks et al., 2014*), are threatened by overfishing, coastal development, eutrophication, and climate change (*Steneck et al., 2002*; *Orth et al., 2006*). Species that have a large capacity to ameliorate impacts of ocean acidification are potential candidates for conservation or restoration. In general, efforts to conserve and restore coastal macrophytes are important for maintaining this capacity in the face of predicted climatic changes (*Nielsen et al., 2018*).

## ACKNOWLEDGEMENTS

We thank P Wallingford for assistance in the field and K Monuki for help with sample preparation and processing in the lab. Logistical support was provided by K Brown, T Hill, S Olyarnik, and J Sones at Bodega Marine Laboratory. Comments from J Ruesink and two anonymous reviewers substantially improved this manuscript. This is CSUN Marine Biology contribution #269.

### Funding

This research was made possible through funding provided by the University of California, Irvine, including a Seed Funding Track 1 award to Cascade Sorte and Matthew Bracken from the Office of Research. The funders had no role in study design, data collection and analysis, decision to publish, or preparation of the manuscript.

### Grant Disclosures

The following grant information was disclosed by the authors:
University of California, Irvine.

### Competing Interests

The authors declare there are no competing interests.

### Author Contributions

- Matthew E.S. Bracken conceived and designed the experiments, performed the experiments, analyzed the data, contributed reagents/materials/analysis tools, prepared figures and/or tables, authored or reviewed drafts of the paper, approved the final draft.
- Nyssa J. Silbiger and Genevieve Bernatchez conceived and designed the experiments, performed the experiments, analyzed the data, contributed reagents/materials/analysis tools, authored or reviewed drafts of the paper, approved the final draft.

- Cascade J.B. Sorte conceived and designed the experiments, performed the experiments, contributed reagents/materials/analysis tools, authored or reviewed drafts of the paper, approved the final draft, and coordinated the project.

### Field Study Permissions

The following information was supplied relating to field study approvals (i.e., approving body and any reference numbers):

This work was conducted with the approval of the California Department of Fish and Wildlife (Scientific Collecting Permit number SCP-13405).

### Data Availability

The raw data are provided in the Supplemental Tables.

### Supplemental Information

Supplemental information for this article can be found online at http://dx.doi.org/10.7717/peerj.4739#supplemental-information.

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
