# Peer review of "Primary producers may ameliorate impacts of daytime CO2 addition in a coastal marine ecosystem"

_PeerJ, doi:10.7717/peerj.4739_

## Round 0.1 · original submission · Minor Revisions

All reviewers agree the work is clearly written, makes a very useful contribution to the field and that it should be published after mainly minor revisions.

Reviewer 1 feels the statistical sections needed some careful rewording, eg clarification of the limitations of the exptl design and the inferences that are drawn.

Reviewer 2 echoes some of the issues raised by Reviewer 1, and suggests the use of alternate statistical techniques eg multiple regression, which the authors may wish to consider, or justify why these were not used.

Reviewer 3 suggests the authors should try to be more circumspect regarding the effect of macrophytes on buffering by a more balanced citing of the relevant literature. This reviewer has noted several references which should be added to the bibliography.

·

Basic reporting

This is a clearly-written and scholarly submission, which sets up the study well in the context of what is known about biological feedbacks to ocean water chemistry from both field and laboratory experiments, as well as natural variability of carbonate chemistry along coastlines. The figures are germane to the story and are internally well-described to a reader.
Issue 1: The only issue that I found with respect to how the manuscript was written applies to the construction of statements regarding relationships among variables, and inference of cause and effect. In my experience, linear regression is applied to a variable (y) that responds to another variable (x), the latter being measured with less error, experimentally manipulated along a gradient, or expected to be a causal factor in the relationship. When one reports the results of a linear regression, characteristically y is related to x. So I found the construction in the manuscript confusing in a couple of places:
Abstract: “Net community production was strongly related to pH, pCO2, net ecosystem calcification, and changes in O2 concentrations.” Accordingly, I would expect to find NCP on the y-axis, and the four other variables each to contribute to variation in NCP. However, I don’t think that this cause-effect relationship (that water chemistry affects NCP) is what the authors intend. An additional point about the construction of this sentence: it is written as if the change over 6 hours is only relevant to a single variable (O2), but results in Fig. 3 emphasize the change in pH and pCO2 as well. It might work, for instance, to write that “under ambient conditions, tide pools with higher NCP also showed larger daytime increases in pH and oxygen, larger declines in pCO2, and higher rates of NEC.”
Methods line 229: The same issue arises here, where four variables are listed (as above, but without O2) but without any indication of which is the response and which the predictor in linear regression analyses, therefore the expectation of cause and effect. So these methods could usefully be rewritten to be clear about which are predictor and which are response variables, as well as to add O2 (as a response variable in relation to NCP, I think – although reduced major axis regression that makes no assumptions about more variation in the y- than x-direction might also work).
Line 260: Same issue – reverse so that the response variables are related to the predictor variable of NCP
Line 285: Same issue – could be rewritten to emphasize not just “relationships among”, but the logical linkage from NCP to improved pH to better conditions for calcification. And on line 288: currently written as “the relationship between x and y”, but typical construction is “the relationship between y and x” (although, again, here as these are two variables expected to be indices of photosynthesis, then their correlation, rather than linear regression results, may be a better approach)

Experimental design

The methods, calculations, and statistical analyses all contribute to a strong test of how biological communities differ in their daytime capacity to draw down carbon (thereby changing pH and pCO2) under the almost-closed field conditions of tide pools at low tide. I was particularly impressed with the attention to best practices in the reporting of carbonate chemistry variables.
Issue 1: Figure 3C, line 229, and line 262: Also worth testing whether NEC is related to the abundance of calcifying organisms. I get that the positive relationship between NEC and NCP is compelling, but a priori I would expect that more calcification occurs when more biomass is present to calcify. Maybe this is tricky because calcification by mussels, Tegula, and coralline algae are not necessarily occurring similarly (but one might also say that about the photosynthesis of the different encrusting and upright fleshy algae…). But I think it would be a relatively quick check, and allow a reader to know that there was not some confounded variable (more primary producers and shelled organisms in some pools than others) driving the relationship in Fig. 3C. In essence, such a test allows a conclusion about whether daytime calcification is driven by per capita rates responding to ambient water chemistry, or to abundance of calcifiers.
Issue 2: There is one last issue in regards to methods, which I will bring up although I don’t think it can really be fixed at this point: how great it would have been to include a tidepool-sized tub of seawater with no benthic organisms present, as a better way of defining the intercept in Fig. 2 (which assumes a linear relationship) AND an excellent proof that the yeast digesters actually added CO2 to the tide pools. That is, since one of the major conclusions of the paper is in regards to variable capacity to ameliorate CO2 addition, depending on NCP, it would be helpful to have samples with very low NCP, as this would confirm the treatment of CO2 addition and help inform that macrophytes are the key players. I personally am satisfied with the conclusions drawn from the existing range of NCP available in the 10 tide pools, but the authors might be a little more circumspect about the certainty of intercept, given the extrapolation beyond data (as in abstract line 31).
Smaller issues:
Line 183: I think that these calculations (NCP, NEC) apply only to tide pools under control treatments. If so, please specify. If not, then I am confused about how it would be possible to infer NCP and NEC from the change in DIC, given that there is an external source when the yeast incubators are bubbling into the tide pools.
Line 188: is a single value used for the density of seawater? I notice that in Silbiger and Sorte 2018, a single value is used, but here the density may change if there is evaporation and salinity changes over 6 hours. I doubt it matters much one way or the other for the calculation, but as written, it is ambiguous.

Validity of the findings

The authors have generally done an excellent job of drawing conclusions from their data. I have already brought up issues of language and experimental design above, which may need small attention in the discussion.
Issue 1: Lines 306 ff are the most speculative, in which daytime drawdown of carbon is stated to improve water chemistry in ways that would more than offset high CO2 and low oxygen from nighttime respiration. I actually think that the authors have missed the most compelling piece of information in this regard, which comes from Silbiger and Sorte 2018 - they show, for some of the same tidepools, that regardless of the fraction of primary producers in the tide pool, nevertheless water chemistry at night converges to a similar low pH. This means that high daytime NCP may not necessarily result in any higher respiration at night, therefore the minimum pH conditions may be rather insensitive to community composition.

Additional comments

The manuscript was a pleasure to read and has definitely stimulated new thinking on my part about the short-term drawdown of carbon by seaweeds.

Reviewer 2 ·

Basic reporting

Please see general comments.

Experimental design

Please see general comments.

Validity of the findings

Please see general comments.

Additional comments

This well written paper describes the effects of macrophytes in mitigating the effects of CO2 addition in Californian tide pools. Although the data are quite limited in their scope, having been collected over a period of 6 hours on two consecutive days, I find the results to be interesting and well-supported. The authors do include some caveats about the limited sampling (only during the day), but I think they should both expand those caveats (only during the spring, only at low tide) and bring them to the front of the paper rather than leaving them to the end. I would also emphasize the caveats around using tidepools, which are really a best-case scenario for identifying these effects, as a proxy for coastal environments writ large.

My other comment is statistical. This dataset strikes me as a good opportunity to use an ANCOVA, multiple regression techniques, or something similar to simultaneously evaluate the effects of CO2 addition and macrophyte abundance. Given the hypotheses presented and the conclusions drawn, one would expect there to be an interaction between CO2 addition and macrophyte abundance such that adding carbon dioxide would have a stronger effect on pH and pCO2 when macrophytes were rare and a weaker effect when they were present. The additional advantage of this approach is that one could partition the variance among these variables. Yes, the authors currently conclude that ignoring the macrophytes causes one to misunderstand how the system is functioning, but it remains unclear to me exactly how important variation in macrophyte abundance really is. Sure, the p-values are significant, but is this a major force in these pools or is it something that only explains a bit more of the variance than one would have explained otherwise? If the authors decide to retain their current statistical approach, I would like to see a justification for not using alternatives such as multiple regressions or ANCOVAs.


Minor comments by line number

51-59: Much of the variability in pH along the coast, in both space and time, is driven by patterns of upwelling. I would be willing to bet a modest sum of money that upwelling dynamics are a more important spatio-temporal forcing variable in the California Current system than local photosynthesis, at least at scales beyond tidepools, semi-contained bays, and other bodies of water with high residence time relative to their volumes. Upwelling deserves at least a mention here as a source of pH variation along the coast. There are several papers by Feely and others to support this.

103-105: In the context of this study, the range of pool volumes and surface areas are at least as interesting as the means, and should also be reported.

122: Please provide the exact dates.

131-133: That seems like a reasonable assumption to me. But how similar were the two days in terms of the weather? Brighter sun on one of the two days could influence photosynthesis, and a stiff breeze could result in increased mixing within the pools and perhaps increased loss of CO2 to the atmosphere.

144: Was there any spatial heterogeneity in carbonate chemistry within the pools? I don’t have a good sense for how quickly CO2 would diffuse or mix out into the entire pool, and if there would be a gradient of CO2 concentration moving away from the air stone.

155: mV is a unit, not a variable. You can say that you measured pH by recording the voltage in mV.

184-190: Although tide pools are closed systems in terms of water at low tide, they are not closed to the atmosphere. Were the pools gaining or losing CO2 across the water-air interface over the course of 6 hours? If so, pool surface area : volume would matter. Is it worth including this as a term in your statistical model, or at least ruling out differences in SA:V between treatments? Or can you justify this effect being so small as to be negligible? Ah, I see you do consider this (line 199 and onward), but it isn’t clear how this was done. You would still need to know the surface area of the pool (not the surface area of the bottom) relative to the volume, and I don’t see that listed a measured variable.

216-217: How was invertebrate biomass calculated? Count data multiplied by some standard for each invertebrate species? Or something else?

234, 273: My natural inclination is to compare Figure 1 to Figure 4 to understand this point. However, Figure 1 is a rate (change per hour), and Figure 4 is an absolute change. I don’t think you need to change the figures, but it would be worth pointing this out so that readers don’t mistakenly assign meaning to the differences in y-axis scaling.

280: Confusing: isn’t the rate of pH change positive for all pools? I assume you mean that the addition of CO2 increased the positive rate of pH change.

281-282: If macrophytes buffer the system by drawing down CO2, how would adding CO2 enhance their ability to draw down CO2? It would be worth trying to provide a potential mechanism for that surprising result.

308: Perhaps daytime benefits to calcification outweigh (unmeasured) night-time costs, although I think you need to state that the effects of macrophytes have not been quantified at night to determine their actual importance. However, reduced calcification is not the only effect of ocean acidification. Couldn’t macrophytes indirectly harm other species at night through physiological effects of elevated CO2?

340: The word “essential” seems overly strong here.

Figure 1: The y-axis is a bit of a mind bend. I would add a second set of y-axis labels to the right hand side of the plots (the frame around the graph already provides a second y-axis on the right), and label the area above zero as “pH increases more rapidly with CO2 addition” and the area below zero as “pH increases less rapidly with CO2 addition”. At least, I think that is what the graph is telling me. A similar statement can be included on panel B for effects on pCO2.

Figure 2: Put a box around your legend so that the reader doesn’t mistake your symbol legends as data points. Or you could move the legend outside of the frame.

Figure 4 legend: The 2 in pCO2 is not subscripted in one instance.

Reviewer 3 ·

Basic reporting

The authors examine the biological and chemical interactions in seawater, using tidepools to look at these interactions in situ. They link tidepool composition to changes in seawater carbonate chemistry within tidepools. The study uses a paired experiment in 10 tidepools where CO2 is added with yeast (n=5), then switching the treatments to examine the effects. The effect of the CO2 appeared more related to the composition of the tidepool (particularly photosynthesizers) than to the treatment per se. The methods, results and figures are generally clear.

Experimental design

The study uses a paired experiment in 10 tidepools where CO2 is added with yeast (n=5), then switching the treatments to examine the effects. While the scatterplots presented seem a little sparse, the message still seems robust and convincing.

Validity of the findings

The study demonstrates the local effects that biological processes of photosynthesis and respiration have on coastal seawater chemistry and is thus worth publishing. Again, the sample size is not great, but the patterns are clear. I have several comments for the authors, but all are relatively minor and can be addressed with the data on hand.

On L176 – Be clear about what was measured versus what was estimated with ‘seacarb’. TA was measured, I gather. However, it was never reported (that I could see), but only used for estimating calcification. The authors should be cautious about using the change in TA to infer calcification because there is evidence that the primary producers change alkalinity through the process of fixing carbon. See e.g. Brewer and Goldman 1976 L&O, Stepien et al 2016 PlosOne. These studies suggest that TA changes are occurring independent of calcification rates and the authors should consider this possibility in their NEC estimate. It is likely the case that their NEC estimates may be confounded with other processes.

Additional comments

Abstract – Sentence “Effects of CO2 additon on pH and pCO2 were not apparent unless we accounted for seaweed surfgrass abundances” is a bit vague.

L 145: the statement ‘due to nonlinearity in the carbonate parameters across time …” seems important. Does this nonlinearity reveal anything about the proposed ‘buffering’?

Fig 4 caption – the phrase “when community composition was not taken into account” is vague. It is simply the result. And in some ways, it seems that this should come first – then the exploration of plant biomass, etc. The SE bars are large and also motivate investigating the composition.

There were several instances where I thought that the authors could have used a more foundational or exact citation:
L 49: Wootton et al. 2008 PNAS, Thomsen et al 2010 Biogeosciences
L 67: The Nielsen et al citation is a non-peer reviewed report and contains no data showing any ’buffering’ and is thus not an accurate citation for the statement. Additionally, the Silbiger and Sorte citation has some of the same data reported here – correct? So I suggest that the authors examine other studies. One suggestion is Semesi et al. 2009 MEPS would be better. Or Delille et al 2000 Polar Biology

In general, the authors should be more rigorous about their citations of macrophyte effects. My take is that there are several papers that show variable carbonate parameters inside of any area with dense macrophytes (Frieder, Koweek), but these do not necessarily show ‘buffering’ and are not always compared with areas without macrophytes, limiting our inference. Just a cautionary statement for all.

---

## Round 0.2 · accepted · Accept

In my re-reading of the ms I wonder if you could double-check the formula for air-sea flux on line 233. Usually this does not have an explicit density term (rho) - it may be correct, just wanted to note that.

Otherwise, this manuscript is Accepted.

#